# Selection, Identification and Functional Performance of Ammonia-Degrading Microbial Communities from an Activated Sludge for Landfill Leachate Treatment [note 1]

**DOI:** 10.3390/microorganisms11020311

**Published:** 2023-01-25

**Authors:** Rossana Petrilli, Attilio Fabbretti, Alex Cerretani, Kathleen Pucci, Graziella Pagliaretta, Matteo Picciolini, Valerio Napolioni, Maurizio Falconi

**Affiliations:** 1School of Biosciences and Veterinary Medicine, University of Camerino, 62032 Camerino, MC, Italy; 2Eco Control Laboratorio Ascolano s.r.l., 63900 Fermo, FM, Italy; 3Eco Elpidiense s.r.l., 63821 Porto Sant’Elpidio, FM, Italy; 4Synbiotec s.r.l., 62032 Camerino, MC, Italy

**Keywords:** biological nitrogen removal, nitrification-denitrification, landfill leachate, activated sludge, microbial community, metagenomics

## Abstract

The increasing amounts of municipal solid waste and their management in landfills caused an increase in the production of leachate, a liquid formed by the percolation of rainwater through the waste. Leachate creates serious problems to municipal wastewater treatment plants; indeed, its high levels of ammonia are toxic for bacterial cells and drastically reduce the biological removal of nitrogen by activated sludge. In the present work, we studied, using a metagenomic approach based on next-generation sequencing (NGS), the microbial composition of sludge in the municipal wastewater treatment plant of Porto Sant’Elpidio (Italy). Through activated sludge enrichment experiments based on the Repetitive Re-Inoculum Assay, we were able to select and identify a minimal bacterial community capable of degrading high concentrations of ammonium (NH_4_^+^-N ≅ 350 mg/L) present in a leachate-based medium. The analysis of NGS data suggests that seven families of bacteria (Alcaligenaceae, Nitrosomonadaceae, Caulobacteraceae, Xanthomonadaceae, Rhodanobacteraceae, Comamonadaceae and Chitinophagaceae) are mainly responsible for ammonia oxidation. Furthermore, we isolated from the enriched sludge three genera (*Klebsiella* sp., *Castellaniella* sp. and *Acinetobacter* sp.) capable of heterotrophic nitrification coupled with aerobic denitrification. These bacteria released a trace amount of both nitrite and nitrate possibly transforming ammonia into gaseous nitrogen. Our findings represent the starting point to produce an optimized microorganisms’s mixture for the biological removal of ammonia contained in leachate.

## 1. Introduction

The economic development and urbanization which occurred, during the last century, in Italy and in all developed countries, led to the production of increasing amounts of municipal solid waste (MSW) (reviewed in the Ref. [1]). Due to the Italian political choice to strongly promote differentiated collection, only a small fraction of MSW has been incinerated ([2] and references therein) and waste incineration plants are practically absent in central and southern Italy. Thus, because of low cost and easy management, MSW are mostly treated in landfill generating a very typical waste, the leachate. 

Leachate is a dark-brown and bad-smelling liquid formed in landfills by rainwater percolation through solid wastes. Generally, landfill leachate is characterized by the presence of elevated concentrations of ammonia and organic compounds in addition to other toxic elements, such as heavy metals (reviewed in the Refs. [3,4,5]). The quality of landfill leachate frequently varies, being affected by many factors (i.e., composition of MSW solid, abundance of rainfall and climate), thus it is constantly monitored using such chemical parameters as pH, chemical oxygen demand (COD), biochemical oxygen demand (BOD), chloride ions (Cl^−^) and ammonium nitrogen (NH_4_^+^-N). Currently, the most used classification of leachate is based on landfill age: young (less than 5 years old), middle (5–10 years old), and mature (more than 10 years old) (reviewed in the Refs. [5,6]). Young leachate shows high values of BOD and COD with a BOD/COD ratio ranging from 0.5 and 1. During aging of this pollution (older leachate), the BOD and COD parameters decline whereas huge amounts of ammonia accumulate (2000–4000 mg/L). This classification is mostly applicable for leachates produced in landfills located in warm temperature zones whereas leachates from equatorial and tropical regions are subjected to a very fast aging and more frequently contain recalcitrant compounds and inhibitors of biological reactions [7,8,9]. Because of the detrimental impact of ammoniacal nitrogen (NH_4_^+^), all the regulatory agencies request the industries, factories and municipals to treat their effluents for ammonia to meet the requirements. Italian Regulation No. 152/2006 established the limits for ammonia wastewater content discharged into shallow water ≤ 15 mg/L and into sewage network ≤ 30 mg/L. The limit for nitrite in the effluent is 0.6 mg/L whereas for nitrate it is 20 and 30 mg/L in shallow water and the sewage network, respectively. 

Many physico–chemical methods involving adsorption, struvite precipitation, coagulation/flocculation, air-stripping and filtration were developed to treat landfill leachate [5,6,10,11,12]. However, the biological removal of complex organic matter and nitrogen by the microbial community forming the activated sludge remains the most widely used and somewhat the most promising approach [3,4,5,13]. This process occurs in conventional wastewater treatment systems, both in aerobic and anerobic conditions. Thus, nitrogen-fixing bacteria convert the nitrogen in the wastewater into ammonia (NH_3_) which during the nitrification process is oxidized to nitrite (NO_2_^−^) by ammonia-oxidizing bacteria (AOB). Subsequently, the NO_2_^−^ is rapidly oxidized to nitrate (NO_3_^−^) by nitrite-oxidizing bacteria (NOB) and finally, in the denitrification process, NO_3_**^−^** is reduced to the gaseous nitrogen form which can directly escape into the atmosphere [14,15,16,17,18]. In particular, mature leachate creates serious disposal problems because of its high content of ammonia associated with a deficiency of biodegradable carbon sources utilized by bacteria of the activated sludge [19]. In fact, high levels of ammonia (NH_3_) are toxic for bacterial cells and dramatically reduce, in municipal wastewater treatment plants (MWTPs), biological nitrogen removal by the activated sludge. 

In the present study, we investigated by a metagenomic approach consisting of next-generation sequencing (NGS) [20] of 16S rRNA genes, the composition of sludge microbes in the municipal wastewater treatment plant (WWTP) of Porto Sant’Elpidio (FM, Italy) with the aim of increasing bioreactor performance. This WWTP, located in central Italy near the Adriatic coast and managed by the Eco Elpidiense s.r.l. company, serves about 50,000 inhabitants that can increase in summer for tourist arrivals. In addition, the Porto Sant’Elpidio WWTP collects and treats a huge quantity of leachate, after being pre-treated via physico–chemical processes, from numerous landfills in a large surrounding area. This implies a very big load in terms of ammonia. Thus, the structure of the microbial community of the native sludge from Porto Sant’Elpidio WWTP was compared to those selected after multiple ammonia stresses in leachate-based medium. This allowed us to identify those microorganisms which can survive at elevated NH_4_^+^ concentrations as those present in leachate and exhibit a high ammonium removal capacity.

Our results indicate that such a strong selection drastically reduced the number (usually more than 150) of different bacterial species, causing a considerable enrichment of certain genera as a function of ammonia stress prolongation. In particular, *Haliscomenobacter*, *Leadbetterella* and *Stenotrophomonas* were predominant (10–30%) after 10–15 days to be then replaced by *Rhodanobacter*, *Castellaniella* and *Nitrosomonas* (10–40%) in 75-day cultures. Therefore, three isolates, belonging to *Klebsiella* sp., *Castellaniella* sp. and *Acinetobacter* sp. genera were further tested for their capability to perform combined heterotrophic nitrification and aerobic denitrification.

## 2. Materials and Methods

### 2.1. Description of Porto Sant’Elpidio Municipal Wastewater Treatment Plant 

The municipal WWTP is placed about 500 meters from the Adriatic Sea coast on Porto Sant’Elpidio (FM, Italy). It has a maximum treatment capacity of a 62,000 population equivalent, and besides, Porto Sant’Elpidio serves some nearby towns. The WWTP, schematically represented in Figure 1, is composed of four different reaction basins of a volume of 1450 m^3^. In these basins, nitrification and denitrification processes occur alternatively and they are controlled by specific probes which measure the quantity of oxygen dissolved in the active sludge. The overall depurative process includes three consecutive steps: (i) the physico treatment; (ii) the biological treatment; and (iii) the final chemical treatment. The wastewater reaches the plant and bar screens are used for removal of coarse solids. Next, sands are removed by gravity sedimentation. After that, the biological treatment occurs through the use of an activated sludge, which in anoxic conditions, favors the reduction of nitrogen and consequently the consumption of organic matter, thus greatly decreasing the nutrient contents in the wastewater. The resulting effluent displays a much lower organic and nitrogen content, with an average removal efficiency of about 95% from the initial values of both COD and nitrogen values. Before being discharged, the wastewater gets a chemical disinfection by a peracetic acid that produces wastewater with a reduced microbic population. In addition, this WWTP treats a large quantity (~100,000 tons per year) of leachate, coming from numerous landfills in a large area of central Italy, which results in about 12,000 equivalent people in terms of organic matter and nitrogen. The quality of four characteristic landfill leachates commonly treated in Porto Sant’Elpidio WWTP is reported in Appendix A. Initially, a mixture of these leachates is subjected to physico–chemical treatments in two separate plants, including coagulation/precipitation, filtration on active carbon and adsorption/ion exchange. These pre-treatments reduce the ammonia level from 1500–2000 mg/L to approx. 500 mg/L. Then, leachate after dilution (~1:100) with the urban wastewater is released into the plant where an additional biological reduction of ammonia occurs (Figure 1). The average composition of pre-treated leachates, used in our experiments, is shown in Table 1.

### 2.2. Media and Cell Growth Conditions 

Aliquots of activated sludge, as indicated in *Figure Legends*, were diluted in Leachate Minimal Medium (LMM) and incubated in a rotary shaker at 28 °C. The LMM was prepared by combining 60 mL of leachate with 30 mL of MM medium and 90 mL of distilled water. The MM broth is a medium originally used to grow *N. europaea* (ATCC #2265) and is composed of three solutions to be mixed together. Solution 1: 4.95 g (NH_4_)_2_SO_4_, 0.62 g KH_2_PO_4_, 0.27 g MgSO_4_, 0.04 g CaCl_2_, 0.5 mL FeSO_4_ (30 mM in 50 mM EDTA at pH 7.0), 0.0002 g CuSO_4_, 1.2 L distilled water. Solution 2: 8.2 g KH_2_PO_4_, 0.7 g NaH_2_PO_4_, 3 L distilled water, bring to pH 8.0. Solution 3: 0.6 g Na_2_CO_3_, 12 mL distilled water. The three solutions were sterilized by filtering and leachate by autoclave. The ammonia concentration of bacterial cultures was adjusted to the indicated values adding (NH_4_)_2_SO_4_ (1 M). Agar plates were prepared using leachate minimal medium (LMM) supplemented with leachate and agar (1.7%) at a final concentration of NH_4_^+^ of 350 mg/L. 

### 2.3. Analytical Methods

The ammonium concentration was estimated by Nessler’s assay carried out in a total volume of 25 mL using 200 μL of each culture and 0.5 mL of Nessler’s reagent. Ammonium rates were measured spectrophotometrically at a wavelength of 410 nm.

Nitrites and nitrate concentrations were determined by using a Dionex ICS-1100 (ThermoFisher Scientific, Waltham, MA, USA) ion chromatography system equipped with a DRS 600 suppressor and a conductivity detector. Anions were separated by a Dionex IonPac AS23 column and a Dionex IonPac AG23 guard column with a flow rate of 1 mL/min of a 0.45 M Na_2_CO_3/_0.08 M NaHCO_3_ eluent. The determinations were conducted according to the APAT CNR IRSA 4020 Man. 29/2003 method, published by the Italian Environmental Protection Agency. The analytical procedure was conducted under UNI CEI EN ISO/IEC 17025:2018 standards; therefore, extensive method validation and the expanded uncertainty were available. Detection limits for nitrites (as NO_2_^−^) were 0.09 mg/L and 3.4 mg/L for nitrates (as NO_3_^−^). 

Total nitrogen was determined by using the small-scale sealed-tube kit (LCK 138, Hach company, Loveland, CO, U.S.A.). Nitrogen compounds present in the samples were oxidized to nitrate according to the method EN ISO 11905-1:1998 which uses peroxodisulfate and a high temperature (120 °C for 30 min) for the digestion. Next, a solution of 2,6-Dimethylphenol was added to the sample, which reacts with the nitrates to form 2,6-Dimethyl-4-nitrophenol. The formed nitrophenol was then determined spectrophotometrically at a wavelength of 345 nm. 

COD was determined by using the sealed-tube test method (ISO 15705:2002), while the BOD was determined by using a small-scale sealed-tube kit (LCK 555, Hach company, Loveland, CO, USA). 

Chlorides were determined by titration with silver nitrate (APAT CNR IRSA 4090 Man 29:2003) and metals were determined after microwave-assisted aqua regia digestion (UNI EN ISO 15587-1:2002) with an ICP-MS (UNI EN ISO 17294-2:2016).

All the analytical procedures described were performed at the Eco Control Laboratorio Ascolano s.r.l. (Italy), (Certification UNI CEI EN ISO/IEC 17025:2018).

### 2.4. Analysis of Microbial Communities by 16S rRNA Gene Next-Generation Sequencing

Bacterial cells from native sludge and samples of Repetitive Re-Inoculum Assay (RRIA) (aliquots of 10–20 mL) were harvested by centrifugation (8000 rpm for 20 min). Chromosomal DNA was extracted by the E.Z.N.A. Kit (Omega Bio-tek Inc., Norcross, GA, USA) according to the instructions given by the manufacturer. 

DNA concentration was estimated by NanoDrop (ThermoFisher Scientific, Waltham, MA, USA). The V3-V4 hypervariable regions of 16S rDNA were amplified using universal primers (341F 5′-TCGTCGGCAGCGTCAGATGTGTATAAGAGACAGCCTACGGGNGGCWGCAG-3′, 805R 5′-GTCTCGTGGGCTCGGAGATGTGTATAAGAGACAGGACTACHVGGGTATCTAATCC-3′) following the 16S Metagenomics Sequencing Library preparation protocol [21]. Libraries were sequenced using the MiSeq Illumina Platform (Illumina Inc., San Diego, CA, USA) with a 2 × 250 paired-end run. Poor quality reads were filtered with Trimmomatic [22]; paired-end reads were merged using FLASH [23] and processed with VSEARCH [24] to detect potential chimera sequences and to cluster merged amplicons in operational taxonomic units (OTUs), with a minimum pair-wise identity threshold of 97%. The NCBI 16S RefSeq database [25] was employed for taxonomic classification. Evaluation of microbial alpha (Alpha-diversity, Chao1, Simpson’s and Shannon’s diversity) and beta (UniFrac distances, Bray–Curtis dissimilarity) diversity measures were performed using an internal pipeline. 

### 2.5. Isolates Identification by 16S rRNA Gene Sanger Sequencing

Cells from isolated colonies were picked up with a sterile loop and directly dissolved in the PCR reaction mix for 16S rRNA gene amplification using the forward primer #838F (5′-AGAGTTTGATCMTGGCTCAG-3′), reverse primer #839R (5′-TACGGYTACCTTGTTACGACTT-3′) and the high-fidelity DNA polymerase (Ex Taq, Takara, Bio Inc., Shiga, Japan). PCR steps were: DNA denaturation at 94 °C for 30 s, primer annealing at 59 °C for 30 s and extension at 72 °C for 90 s (28 cycles). The amplification products were purified by the Real Genomics kit (RBC Bioscience Corp., New Taipei, Taiwan) and sequenced by the Sanger method at BMR Genomics (Padova, Italy). Finally, sequences were analyzed using Nucleotide BLAST (https://blast.ncbi.nlm.nih.gov/Blast.cgi) accessed on 1 March 2022 

## 3. Results and Discussion

### 3.1. Nitrogen Removal Rate by Activated Sludge under Ammonia and Salt Stresses

This study was promoted by the Eco Elpidiense s.r.l., a private company that manages the municipal wastewater treatment plant of Porto Sant’Elpidio (Figure 1). This WWTP, besides its routinely depuration activity, is overloaded by the disposal of large amounts of landfill leachate coming from a vast region of central Italy. Although after high dilutions, the spillage of big volumes of leachate into the municipal wastewater treatment plant represents a relevant risk. In fact, the reactor has to withstand peaks of ammonium (NH_4_^+^-N ≅ 100–150 mg/L) that could be toxic for bacteria of the sludge, thereby resulting in a slowing down of or even stopping the biological nitrogen removal. Even though this is a challenging condition, the Porto Sant’Elpidio WWTP has a mean nitrogen removal rate of about 95%, as determined by the routinely analyses carried out at the Eco Control Lab, Ascolano (personal communication). This suggests that microorganisms of the active sludge adapted to these continuous NH_4_^+^ stresses make this plant very attractive for the analysis of its microbial community. 

Thus, according to the Eco Elpidiense s.r.l. requirements, the primary aim of this study was to investigate and eventually to improve the WWTP performance in terms of the biological conversion of ammonia to nitrite, known also as partial nitrification or nitritation [26]. This reaction is certainly the critical one of the conventional nitrification–denitrification process by bacteria, and it is nevertheless the most energy-consuming step due to the aeration system which injects air into the plant. Thus, the microbial community taken from the Porto Sant’Elpidio WWTP was tested for its ability to tolerate ammonia stress in order to estimate a threshold value at which bacteria were still able to efficiently eliminate ammoniacal nitrogen. For this purpose, bacteria from the sludge were grown in minimal broth (LMM) supplemented with only leachate and containing increasing concentrations of ammonia. The results presented in Figure 2A,B revealed that good ammonia degradation was maintained for the two lower NH_4_^+^-N concentration curves (250 and 400 mg/L) where nitrogen removal rates were 70–85% and 50–75% at 3 and 8 days, respectively. Conversely, the nitrification process considerably slowed down and was dramatically inhibited for higher amounts of NH_4_^+^ (600–900 mg/L). Thus, this assay indicated that microorganisms of native sludge well-tolerate ammonium stress and retain high nitrification activity up to ~400 mg/L of NH_4_^+^-N. This finding was very promising, considering the experimental conditions used. In fact, to the best of our knowledge, most previous studies estimated the biological ammonia removal rate in wastewaters containing low NH_4_^+^-N (≤100 mg/L). For this reason, data comparison is quite complicated. 

Instead, Jiang et al. [27] used an elevated NH_4_^+^ concentration (300 mg/L) in a laboratory-scale reactor where the nitrification process reached about 94%. Differently from our experiments, carried out in common microbiology 150 mL flasks, this reactor was not a closed system, and to feed bacteria, the COD level was maintained constant (500–600 mg/L) for ~50 days by adding external organic compounds. Remarkably, the native microbial community of our sludge showed excellent nitrification performance (~85%) metabolizing, as only a carbon source, the organic materials in leachate without the need of extra nutrients. In addition, due to the fact that landfills, in the last decade, receive pre-treated waste (i.e., separate waste collection), the leachate generally contains low quantities of putrescible matter and is characterized by a low BOD/COD ratio. According to this, the leachate we used in our experiments had a COD/BOD ratio ≤ 0.3 (Table 1), thereby being poorly biodegradable and difficult to be treated by the conventional nitrification-denitrification process [28]. Under these experimental conditions and in the presence of oxygen, the nitrite was the main product of ammonia oxidation (Figure 2C), whereas the nitrate formation was negligible. In fact, the maximum concentration of NO_3_^−^-N for the four different cultures ranged from 4 to 8 mg/L (data not shown). 

Landfill leachate, managed by the WWTP of Porto Sant’Elpidio, contains high amounts of chloride ions (Cl^−^) that, depending on the local climate (rainfall and temperature variations), range from 2 to 4 g/L as analytically determined by the Eco Control Laboratory (FM, Italy). In addition to ammonia, an elevated Cl^−^ content represents a further problem for leachate treatment in WWTPs. Thus, the effect of salt stress on the ability of bacteria to degrade ammonia was investigated. As seen in Figure 3, concentrations of Cl^−^ below 2 g/L slightly affected the NH_4_^+^ removal rate, whereas it was reduced to 50% at 4 g/L of chloride ions. For Cl^−^ values greater than 6 g/L, the pollutant degradation by the bacterial population of the sludge was strongly impaired.

### 3.2. Repetitive Re-Inoculum Assay

In this study, we devised a simple experiment, named the Repetitive Re-Inoculum Assay (RRIA), somewhat resembling the dilution to extinction approach [29,30] in which an uneven microbial community from environmental samples was subjected to serial dilutions. These previous studies [29,30] demonstrated that important changes, depending on dilution, were found in the community structure and in functional characteristics between the original and reformed communities. 

RRIA consists of consecutive re-inocula/dilutions, of the same sludge culture, in leachate-based medium (LMM) with an elevated NH_4_^+^-N content. The primary aim of RRIA was to reduce the complexity of the bacterial population by selecting those species able to tolerate repeated ammonia stresses and characterized by high efficiency in nitrogen removal. Monitoring the ammonia level, this assay gives rise to a peaks-and-falls plot, as shown in Figure 4A. Basically, a representative RRIA was performed as follows: (i) bacteria from activated sludges were inoculated in a liquid minimal medium supplemented with only leachate and grown at 28 °C; (ii) ammonia concentration was estimated at the starting point (first inoculum) and at regular time intervals for the next 75 days; (iii) as bacteria started to degrade ammonia and its concentration dropped to about 50 mg/L (falls), about one-fourth of the culture volume was re-inoculated in leachate medium so that the initial value of NH_4_^+^-N (~350 mg/L) was restored (peaks); (iv) this step was performed over and over again (seven times in RRIA of Figure 4A) during the two months of the experiment, hence the name “Repetitive Re-Inoculum Assay”; (v) aliquots of the culture were withdrawn at different falls of the ammonia level curve (NH_4_^+^-N ≅ 50 mg/L in Figure 4A) and cells of bacterial communities were harvested to be investigated (see below). Comparable NH_4_^+^ curves were obtained in additional RRIAs where the ammoniacal nitrogen concentration of the starting point and of each re-inoculum was set to 150 and 220 mg/L (Appendix A). 

Unlike the NH_4_^+^ stress experiments (Figure 2) where the NO_3_^−^ formation was negligible, a complete ammonia oxidation took place in RRIA, resulting in accumulation of both nitrites and nitrates (Figure 4B). This observation suggests that selection by RRIA modified the structure of the native bacterial community changing relative abundance, proliferation and nitrogen removal activity of nitrifying AOB, NOB and possibly complete ammonia oxidation (comammox) bacteria [31,32,33]. Notably, RRIA was prolonged for 75 days, ~10 times longer than the experiment of Figure 2, and it was not a closed system. In fact, the same culture was repeatedly inoculated in fresh leachate medium assuring adequate levels of organics, thus favoring, depending on nutritional requirements, the predominance of some species over others (see metagenomic analysis).

Two microbial communities corresponding to the 27th and 53th days of RRIA were compared with the native sludge for their efficiency to degrade ammonia in leachate minimal medium. As shown in Figure 4B, the G5-2 and G5-4 bacterial populations almost completely oxidized ammonia after 9–11 days, while within the same time interval, the control culture derived from the native sludge still contained 200 mg/L of NH_4_^+^-N. Notably, this experiment was carried out at a considerably high ammonium concentration (~370 mg/L) explaining the low removal rate observed (compare to Figure 2A at similar NH_4_^+^-N concentration). 

### 3.3. Determination of Ammonia-Degrading Microbial Communities by 16S rRNA Amplicon NGS

Selected samples from RRIA, collected at 0, 10, 15 and 75 days, were analyzed through 16S rRNA NGS. The total number of raw and post-processed sequenced reads is reported in Table 2. The total number of obtained raw reads ranges from 98,151 to 18,060. While the samples collected at 0, 10 and 15 days were quite homogeneous regarding the sequencing/alignment metrics (Table 2), the sample collected at 75 days showed a quite small number of sequenced reads when compared to the other samples. Nonetheless, rarefaction curves (Appendix A) demonstrated that the sample collected at 75 days also reached the maximum number of mappable OTUs. This is clearly attributable to the reduced microbial community selected at 75 days through the RRIA approach. Alpha-diversity metrics, reported in Table 3, show that the sample collected at 75 days displays, as expected, an important drop-down in all the alpha-diversity metrics. The analysis of beta-diversity, determined by Principal Coordinate Analysis (PCA) of both UniFrac distance and Bray–Curtis dissimilarity (Figure 5), further highlights the extremely different microbial composition of the sample collected at 75 days compared to the other three time-points.

Conversely, seven different families (Alcaligenaceae, Nitrosomonadaceae, Caulobacteraceae, Xanthomonadaceae, Rhodanobacteraceae, Comamonadaceae and Chitinophagaceae) showed a progressive increase in frequency from time-point 0 to time-point 75 (Figure 6A). Notably, the Rhodanobacteraceae increased from 1% at time-point 0, to 40.2% at time-point 75. Some of the families (Cytophagaceae, Marinilabiliaceae, Sphingomonadaceae, Rhodobacteraceae, Haliscomenobacteraceae and Flavobacteriaceae) did not display a linear increasing/decreasing trend across the four time-points. However, the Haliscomenobacteraceae became the prevalent bacteria both at time-point 10 (32%) and at time-point 15 (18.1%), thus disappearing at time-point 75. Similarly, the Cytophagaceae became the second most prevalent bacteria at time-point 15 (15.8%) while disappearing at time-point 75.

When considering the composition of the microbial communities at the genus level, we identified 14 different genera with a cut-off of 5% in at least one sample (Figure 6B). *Brevundimonas* and *Nitrosomonas* showed a progressive increase across the four time-points, starting from being nearly absent at time-point 0, to peaking at 5% and 9.6%, respectively, at time-point 75. Conversely, *Geobacter*, *Thauera* and *Zoogloea* that represent the main bacteria at time-point 0 (6.4%, 9.5%, 29.8%, respectively), progressively disappeared across time-points 10, 15 and 75. The other genera did not show trends, although *Haliscomenobacter* represented the main genus at both time-points 10 (31.8%) and 15 (18%). It should be noted that *Leadbetterella* and *Stenotrophomonas* represented the second (13.6%) and the third (9.3%) most prevalent bacteria at time-point 15, respectively. Notably, *Rhodanobacter, Castellaniella* and *Nitrosomonas* became the predominant bacteria at time-point 75, with percentages of 40.2%, 14.3% and 9.6%, respectively. 

The analysis of bacterial communities suggested that a concomitant autotrophic/heterotrophic nitrification took place during the Repetitive Re-Inoculum Assay. Our results are consistent, in terms of identified taxa, with those recently obtained by Wang et al. [34] who studied biological nitrogen removal in a pilot-scale reactor for six months. Thus, during the initial part of the RRIA (first 15 days), NH_4_^+^ removal was promoted by heterotrophic AOB mainly belonging to the Xanthomonadaceae, Cytophagaceae and Chitinophagaceae families. Heterotrophs do not oxidize ammonia as an energy source and by virtue of their ability to use the organic matter contained in leachate, they are characterized by a high growth rate and strong resistance to extreme NH_4_^+^ stress (≥300 mg/L) [35]. At genus level, the most abundant was *Haliscomenobacter* (~20–30%), a filamentous bacterium found in high ammonium-loaded activated sludge [36] and belonging to Haliscomenobacteraceae. Other genera as *Leadbetterella* (Cytophagaceae) and *Stenotrophomonas* (Xanthomonadaceae) were part of two largely represented families (~10–15%) as identified in Figure 6A. Chitinophagaceae was the third dominant AOB with a relatively high frequency (~8%), as previously reported by Wu et al. [37] analyzing the microbiota of a pilot sludge digester. In the final part of RRIA, possibly due to their slow growth, autotrophic AOB as *Nitrosomonas* accumulated over time, reaching ~10% in the two months of cultures. This genus includes some of the most important nitrifying species as *N. europaea and N. eutropha* that have been extensively investigated for their key role in wastewater treatment systems [38]. They convert ammonia to nitrite in two steps. In the first oxidation reaction, the NH_3_ is transformed into hydroxylamine (NH_2_OH) by ammonia monooxygenase (AMO). Then, this intermediate product is converted to NO^−^_2_ by hydroxylamine oxidoreductase [39]. Notably, a dramatic increase was observed for *Rhodanobacter* (~40%). This genus was found to contribute to the denitrification process in a bench-scale reactor at a low pH [37]. Accordingly, during ammonia removal (NH_4_^+^ falls in RRIA) a gradual acidification of the medium occurred (pH decreased from 8 to 5), explaining the elevated abundance of this bacterial species. Therefore, the Comamonadaceae family, including many denitrifying species, was also increased [40], whereas comammox bacteria of the *Nitrospira* genus [31,32] were recovered at a very low frequency (≤1%).

### 3.4. Heterotrophic Nitrification and Aerobic Denitrification by Isolates

Aliquots of the culture at different times of the RRIA (53 and 75 days), after adequate dilutions, were plated on Petri dishes to isolate single colonies. These Agar plates contained the minimal medium supplemented with only leachate (LMM), ensuring further selection. Many isolates were inoculated in LMM (255 mg/L of NH_4_^+^-N) to test their ability to remove ammonia, and eventually, its oxidation products (NO_2_^−^ and NO_3_^−^). Then, the most promising bacterial strains were identified at genus level by means of the following experimental approach: (i) PCR amplification of the 16S ribosomal RNA gene; (ii) Sanger sequence of 16S DNA; and (iii) nucleotide BLAST analysis.

The levels of the commonly used nitrogenous compounds were estimated during cell growth for the following identified bacteria: *Klebsiella* sp., *Castellaniella* sp., *Acinetobacter* sp. *Arthrobacter* sp., *Thermomonas* sp. and *Sphingomonas* sp. According to NGS analysis (Figure 6A), *Castellaniella* sp. (Alcaligenaceae)*, Thermomonas* sp. (Xanthomonadaceae) and *Sphingomonas* sp. (Sphingomonadaceae) belong to families that were predominant at time-point 75 of RRIA. All isolated bacteria displayed similar activity in nitrogen removal, and three genera are shown in Figure 7. The total nitrogen (TN) concentration, within the first four to six days of culture, rapidly decreased from 380 to 150 mg/L mostly reflecting the reduction (~50%) of ammonia content and, at lesser extent, the degradation of organic nitrogen compounds contained in leachate. Following the initial prompt decline, the total nitrogen showed a slight increase (10–12%) at the 10th day and its level closely accounted for the sum of the ammonia remained, not biodegradable chemical and cellular macromolecules. Notably, bacteria, albeit slowly, were growing in leachate broth with consequent accumulation of organic nitrogen (i.e., proteins) that was detected as TN, thus partially compensating for the progressive decrease of ammonia. These isolates showed a lower NH_4_^+^ removal rate (~70% after 12 days) than that observed for the native sludge at the same ammonia concentration (250 mg/L, Figure 2A,B), possibly because of the lack of reciprocal feeding interactions among diverse microorganisms. Unlike the complex bacterial population of the sludge (Figure 2C), the ammonia degradation by *Klebsiella* sp., *Castellaniella* sp. and *Acinetobacter* released negligible quantities of both nitrites and nitrates. This finding strongly suggests that these bacteria were able to go through with a combined heterotrophic nitrification and aerobic denitrification (HNADM), where NH_4_^+^ is oxidized to NO_2_^−^ and then converted to gaseous nitrogen products (N_2_O or N_2_) that are finally released into the atmosphere [41,42]. However, we should acknowledge as a limitation the lack of determination of the changes in important physiochemical parameters such as BOD and COD related to the progressive removal of nitrogen compounds by bacteria.

During the last decade, heterotrophic nitrification and aerobic nitrite/nitrate denitrification under aerobic conditions was demonstrated for many genera of fungi and bacteria. HNADMs have been found in various environments retaining their nitrification/denitrification capabilities also under stress conditions as low pH, high ammonia and salty wastewater (reviewed in the Ref. [43]). According to previous studies [44,45,46,47], the genera *Acinetobacter* sp., *Klebsiella* sp. and *Arthrobacter* sp., isolated from the Porto Sant’Elpidio municipal WWTP exhibit considerable interest for their potential in nitrogen removal under aerobic conditions and are currently under further investigation in our laboratory.

## 4. Conclusions

The primary aim of this work was to investigate the biological degradation of inorganic nitrogen from landfill leachate. Initially, we evaluated the ammonia removal rate of the activated sludge from the WWTP of Porto Sant’Elpidio (Italy) identifying the upper limits, toxic for bacteria, of NH_4_^+^ (400 mg/L) and Cl^−^ (4 g/L). Then, the microbial composition of the native sludge, determined by NGS, was compared to that selected after repetitive ammonia stresses (RRIA). Differently from other studies, these experiments were carried out in leachate-based medium not containing an additional source of carbon. NGS analysis demonstrated that a significant enrichment of certain bacterial species takes place as a function of incubation time under ammonia stress. In particular, Rhodanobacter, Castellaniella and Nitrosomonas were the predominant genera after two months of RRIA experiments. Finally, the three isolated strains, *Klebsiella* sp., *Castellaniella* sp. and *Acinetobacter* sp. oxidized NH_4_^+^ to NO_2_^−^ that was then transformed to gaseous nitrogen (N_2_O or N_2_), suggesting that these bacteria perform heterotrophic nitrification coupled with aerobic denitrification.

Our findings represent the starting point to produce an optimized microorganism’s mixture for the biological removal of ammonia contained in leachate. This study could have practical implications possibly to increase, in a short time, the bioreactor performance. In fact, we are setting up a large-scale cultivation of this bacterial mixture to be introduced into a pilot reactor of 2000 L available at the Eco Elpidiense Company, thus enriching the native sludge with bacterial species able to resist and very active in degrading elevated NH_4_^+^ concentrations. These tests will permit to monitor changes in fundamental parameters as nitrogen compounds, pH, COD, and BOD with respect to the microbial population composition under experimental conditions that closely resemble those of a real wastewater treatment plant.

## Figures and Tables

**Figure 1 microorganisms-11-00311-f001:**
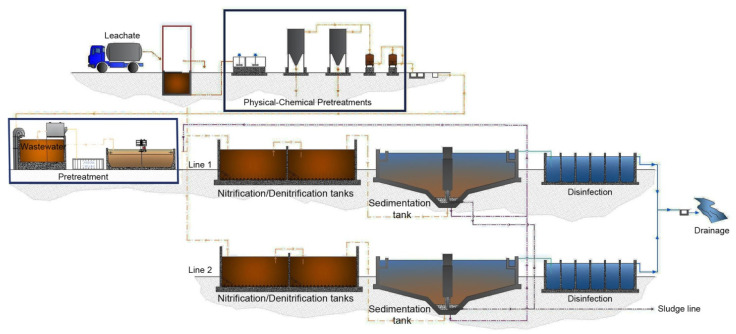
Flow chart of Porto Sant’Elpidio WWTP. A detailed description of the WWTP is provided in Section 2.1.

**Figure 2 microorganisms-11-00311-f002:**
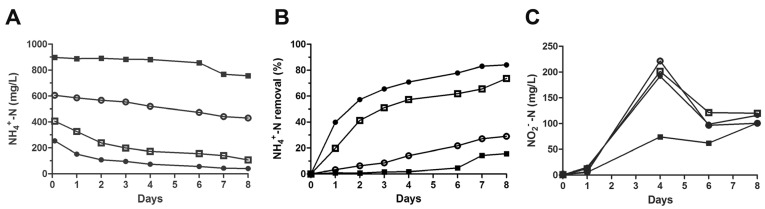
Effects of ammonia stress on NH_4_^+^ removal rate by the activated sludge. Aliquots (10 mL) of the activated sludge (150 mg, dry weight) were mixed with 20 mL of LMM and incubated at 28 °C. Increasing ammonium concentrations, expressed as NH_4_^+^-N, were adjusted in the four cultures to 250 (●), 400 (□), 600 (○) and 900 (■) mg/L and monitored, using Nessler’s test, as a function of time (**A**) and reported as removal percentage (**B**). Nitrite concentration, expressed as NO_2_^−^-N, is reported (**C**). Data represent the average of three independent experiments and the standard deviation is ~10%.

**Figure 3 microorganisms-11-00311-f003:**
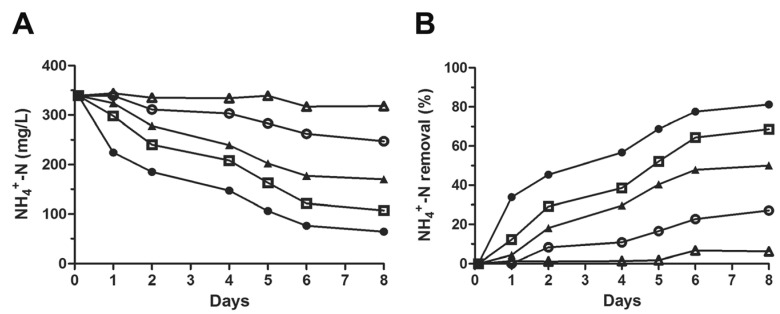
Effects of salt stress on NH_4_^+^ removal rate by the activated sludge. The activated sludge was incubated in LMM essentially as described in the legend of Figure 2. Chloride ion (Cl^−^) concentrations were adjusted to 0.5 (●), 2 (□), 4 (▲), 6 (○) and 10 (Δ) g/L and NH_4_^+^-monitored, using Nessler’s test, as a function of time (**A**) and expressed as a removal percentage (**B**). Data represent the average of three independent experiments and the standard deviation is ~10%.

**Figure 4 microorganisms-11-00311-f004:**
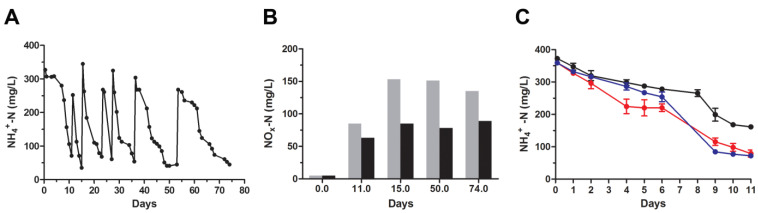
Repetitive re-inoculum assay. (**A**) The activated sludge (10 mL), corresponding to 150 mg dry weight, was mixed with 30 mL of LMM essentially as described in the legend of Figure 2. The culture was placed in a water bath shaker at 28 °C and ammonia concentrations were estimated using Nessler’s test, at the starting point (indicated with day 0) and at regular intervals for the next 75 days. At each NH_4_^+^ fall, 10 mL of the culture was re-inoculated in fresh LMM (30 mL) and ammonia concentration was adjusted over again to ~350 mg/L (peaks). (**B**) Samples from RRIA were withdrawn at NH_4_^+^ falls to determine nitrite and nitrate and their values, expressed as NO_2_^−^-N (grey bars) and NO_3_^−^-N (black bars), respectively, have been plotted. Data represent the average of three independent experiments and the standard deviation is ~10%. (**C**) The curves of the NH_4_^+^ removal rate in LMM for sludge (●) and G5-2 (27 days, ●) and G5-4 (53 days, ●) communities are shown. The initial cell number of the cultures was ~5 × 10^6^ cells/mL as determined by plating bacteria on Luria–Bertani solid medium after serial dilutions.

**Figure 5 microorganisms-11-00311-f005:**
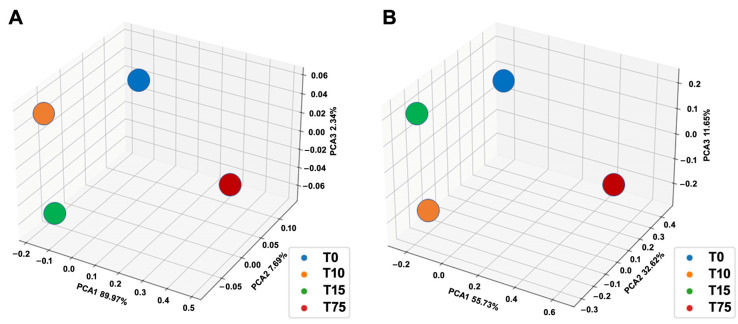
Principal coordinate analysis (PCA) of beta-diversity metrics for the RRIA samples sequenced by 16S rRNA amplicon NGS. (**A**) Unweighted Unifrac distance; (**B**) Bray–Curtis dissimilarity.

**Figure 6 microorganisms-11-00311-f006:**
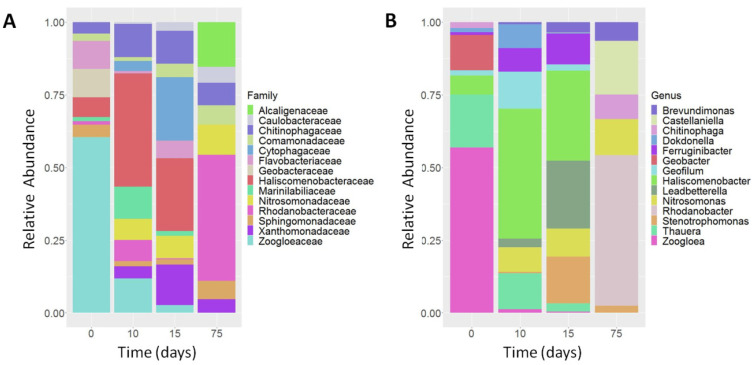
A 100% stacked bar-chart. Changes in the microbial composition, at family (**A**) and at genus (**B**) level of four times during the period of 75 days of RRIA (Figure 4A), as determined by sequencing of 16S rRNA.

**Figure 7 microorganisms-11-00311-f007:**
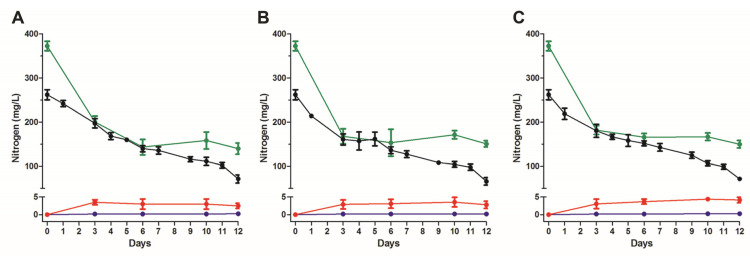
Change of nitrogen compounds’ concentration of isolated bacteria. Bacterial strains were inoculated in LMM (A_600 nm_ = 0.1) and the levels of total nitrogen (●), ammonia (●), nitrite (●) and nitrate (●) monitored during the cell growth at 28 °C. Cultures were: *Klebsiella* sp. (**A**), *Castellaniella* sp. (**B**) and *Acinetobacter* sp. (**C**). Total nitrogen is the sum of ammonia, nitrites, nitrates and organic nitrogen compounds from leachate and cellular macromolecules. The optical density (A_600 nm_) of cultures was 0.3–0.4 at the 12th day.

**Table 1 microorganisms-11-00311-t001:** Average composition of pre-treated leachates used in our experiments.

Parameter	Value ± SD
pH	7.6 ± 0.25
COD (mg/L)	4066 ± 830
BOD (mg/L O_2_)	1207 ± 177
BOD/COD	0.30 ± 0.03
NH_4_^+^-N (mg/L)	1415 ± 166
NO_3_^−^ -N (mg/L)	1.5 ± 1.2
NO_2_^−^ -N (mg/L)	traces
Cl^−^ (mg/L)	3253 ± 472
Cu (mg/L)	0.061 ± 0.04
Pb (mg/L)	0.005 ± 0.01
Cr (mg/L)	1.04 ± 0.53
Ni (mg/L)	0.234 ± 0.12
Zn (mg/L)	0.106 ± 0.11

**Table 2 microorganisms-11-00311-t002:** Sequencing read processing results. Sequencing and alignment metrics for the four selected sequenced samples are reported, from the number of initial raw reads to the number of reads not aligned to the reference NCBI 16S RefSeq database.

Sample	Raw Reads	Trimmed Reads	%_Trimmed_Reads_Over_Raw	Assembled Reads	%_Assembled Reads over Trimmed	Hit Reads	% Hit over Assembled	No Hit Reads
0 days	98,151	91,281	93	50,242	55	22,641	45	27,601
10 days	82,295	76,807	93	46,538	61	30,319	65	16,219
15 days	95,210	88,113	93	53,764	61	33,086	62	20,678
75 days	18,060	16,920	94	12,487	74	7861	63	4626

**Table 3 microorganisms-11-00311-t003:** Alpha-diversity metrics for the RRIA samples sequenced by 16S rRNA amplicon NGS. Alpha-diversity metrics (Shannon’s, Simpson’s, Chao1 and alpha-div) for the four selected sequenced samples are reported.

Sample	Shannon	Simpson	Chao1	α-Diversity
0 days	4.808	0.937	87.5	80
10 days	4.255	0.872	94.667	90
15 days	5.104	0.943	107.333	98
75 days	3.264	0.807	55.5	33

## Data Availability

Not applicable.

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
