# Peer review of "Selection, Identification and Functional Performance of Ammonia-Degrading Microbial Communities from an Activated Sludge for Landfill Leachate Treatmentâ€"

_microorganisms, 2023, doi:10.3390/microorganisms11020311_

Round 1
Reviewer 1 Report
The text that is the subject of this review is devoted to the issue of leachate treatment from landfills.
The treatment of such effluents is a complicated and demanding process due to the high pollutant loads, and developing techniques for treating such effluents is highly desirable. Therefore, this subject is important and shows excellent scientific and practical potential.
The authors presented in the text the results of research on the long-term process of biological leachate treatment, together with the identification of families of bacteria present in the process.
During the reading of the text, the following remarks arise:
• The paper's title does not cover the entirety of the presented material. Apart from "the identification of ammonia-degrading microbial communities from activated sludge", the article also presents the influence of selected parameters (i.e. salt and ammonia stress) on the nitrogen removal and heterotrophic nitrification and aerobic denitrification processes. Itself identification of ammonia-degrading communities is a relatively small part of the text. The title should be extended with the indicated content.
• The reviewed paper contains neither a summary, conclusions, nor an indication of the areas of use of the presented research. Some excerpts from chapter 3 (e.g. L446-453) should instead be included in chapter 1, describing state of the art.
Formal chapters containing a summary and conclusion should be inserted in the text.
• Authors throughout both ammonia (NH3) and ammonium (NH4+) call ammonia. Chemically (but also physically), they are different form of nitrogen compounds, that should not be confused! Look also: https://crops.extension.iastate.edu/cropnews/2008/04/surface-waters-ammonium-not-ammonia-%E2%80%93-part-1 or http://ie.hach.com/parameters/ammonia
• How to explain the growth of total nitrogen in figures7 (A, B and C) on the 10th day? At this particular point in time, total nitrogen content increases significantly despite a decrease of NH4+, a decrease in NO3- and unchanged NO2-.
• In L39, the authors write about landfill wastewater and leachate. What do the authors mean by the term landfill wastewater?
• In L106, the authors write that "wastewater (...) gets grilled". Do authors mean screening bar? (use of bar screens for removal of coarse solids). Jargon should not be used in scientific texts.
• Some technical editing imperfections remained in the text, such as:
a. MgS04 instead of MgSO4 in L134.
b. Spread-out, hard-to-read text L172 (here, the URL of the source document is better to be moved to the bibliography and cited).
As mentioned, the article contains exciting research results of the landfill leachate treatment process. However, it cannot be published in its current form. The formal part should be rebuilt - the text should be supplemented with a summary and conclusions. Edit the title to reflect the actual content of the article. After indicated reconstruction, the text can be subjected to the review process again.
Author Response
We thank the Reviewers and Editors for the positive evaluation of our research and for their valuable suggestions to improve the manuscript. We hope that it is now suitable for publication in Microorganisms.
All changes in the manuscript are indicated as follows:
Deleted text = struck through words
New text, corrections = red words
Reviewer 1
The text that is the subject of this review is devoted to the issue of leachate treatment from landfills.
The treatment of such effluents is a complicated and demanding process due to the high pollutant loads, and developing techniques for treating such effluents is highly desirable. Therefore, this subject is important and shows excellent scientific and practical potential.
The authors presented in the text the results of research on the long-term process of biological leachate treatment, together with the identification of families of bacteria present in the process.
During the reading of the text, the following remarks arise:
The paper's title does not cover the entirety of the presented material. Apart from "the identification of ammonia-degrading microbial communities from activated sludge", the article also presents the influence of selected parameters (i.e. salt and ammonia stress) on the nitrogen removal and heterotrophic nitrification and aerobic denitrification processes. Itself identification of ammonia-degrading communities is a relatively small part of the text. The title should be extended with the indicated content.
Answer. The title has been modified adding “Functional Performance”.
The reviewed paper contains neither a summary, conclusions, nor an indication of the areas of use of the presented research. Some excerpts from chapter 3 (e.g. L446-453) should instead be included in chapter 1, describing state of the art.
Formal chapters containing a summary and conclusion should be inserted in the text.
Answer. Conclusions have been added.
Authors throughout both ammonia (NH3) and ammonium (NH4+) call ammonia. Chemically (but also physically), they are different form of nitrogen compounds, that should not be confused! Look also: https://crops.extension.iastate.edu/cropnews/2008/04/surface-waters-ammonium-not-ammonia-%E2%80%93-part-1 or http://ie.hach.com/parameters/ammonia
Answer. We thank the Reviewer for this clarification. In order to avoid possible confusion, the text has been revised, accordingly.
How to explain the growth of total nitrogen in figures7 (A, B and C) on the 10th day? At this particular point in time, total nitrogen content increases significantly despite a decrease of NH4+, a decrease in NO3- and unchanged NO2-.
Answer. We agree with the Reviewer that the total nitrogen increases in the interval 6-10 days passing from 150 to 170 mg/L (Fig. 7). However, this rise is quite modest corresponding to about 12%. Notably, cells, albeit slowly, are growing and nitrogen contained in cellular macromolecules accumulates. This contributes to make constant the total nitrogen despite the decrease of ammonia.
This explanation has been added to the text at pag. 12.
In L39, the authors write about landfill wastewater and leachate. What do the authors mean by the term landfill wastewater?
Answer. The mistake has been corrected.
In L106, the authors write that "wastewater (...) gets grilled". Do authors mean screening bar? (use of bar screens for removal of coarse solids). Jargon should not be used in scientific texts.
Answer. We apologize for using jargon, the sentence has been modified.
Some technical editing imperfections remained in the text, such as:
- MgS04 instead of MgSO4 in L134.
Answer. The typo has been corrected.
- Spread-out, hard-to-read text L172 (here, the URL of the source document is better to be moved to the bibliography and cited).
Answer. The URL has been moved to the References section.
As mentioned, the article contains exciting research results of the landfill leachate treatment process. However, it cannot be published in its current form. The formal part should be rebuilt - the text should be supplemented with a summary and conclusions. Edit the title to reflect the actual content of the article. After indicated reconstruction, the text can be subjected to the review process again.
Reviewer 2 Report
The research presented is interesting because it deals with real wastewater and the analysis of microbial communities present in the process of oxidation or degradation of NH4; the metagenomic analyses are interesting and contribute to the knowledge of environmental biotechnology, although the data obtained must be contrasted with the physicochemical characteristics of the wastewater and the sludge used. Likewise, it is important to discuss the presence of some microbial species in the possible routes of the nitrogen cycle, mainly in the biological fixation processes.
The following are observations to be taken into account.
1. In the introduction, there is no evidence of antecedents of other similar works and what differentiates this work from others existing in the literature; it is suggested to include references to other similar works and indicates the contribution of this work to new knowledge.
2. The references in lines 39 and 44 are not clear why [Reviewed in 2-4] is indicated. It is suggested to follow the reference format of the journal; in the document, there are several references in this form. If the journal allows it, ignore this observation.
3. It is suggested to review some keywords (Biological nitrogen removal; Nitrification-denitrification) that can fit into a single one, such as biological nitrogen fixation or some other that represents the previous ones.
4. A physicochemical characterization table is shown in section 2.1, but it does not show how the samples were taken, what sample storage methods were used, and what methods were used to store the samples. What methods were used to determine the contaminant load for each parameter evaluated? It is suggested that this information be included, as well as the standard deviation of each measurement.
5. In line 172, in the metagenomics library, there is a reference to a protocol; it is suggested to look for the bibliographic reference and refer it to the references section. Is the protocol already validated, is it published or referenced in a scientific document, and is it a protocol established by the commercial company?
6. Line 192, it is suggested that the internet link be moved to the references chapter, or according to the editorial norms, it should be referenced according to these guidelines.
7. In the section on Nitrogen removal rate by activated sludge under ammonia and salt stresses, because nitrate measurements were not performed, it would be interesting for the authors to determine the nitrate concentration and determine the complete nitrification process, analyzing the percentage of NH4 oxidation to NO3, given that the microbial community in the activated sludge has microorganisms capable of performing the nitrification process. In this same section, it is suggested to include data on volatile suspended solids (biomass) and COD so that the authors can discuss COD removal and the nitrification process with biomass growth and determine if the increase in NH4 concentration causes a substrate inhibition process in the microbial community.
8. What was the age of the sludge? Were the cell retention time and maturity of the sludge determined compared to the NH4 removal rates? It is suggested to include that information in this section.
9. There is no discussion and information on the physicochemical characteristics of the wastewater; likewise, there is no information on the characteristics of the sludge (sludge volumetric index, age, concentration of volatile suspended solids, etc.). Also, it is important to indicate the aeration conditions, mainly dissolved oxygen concentration. These factors are key to determining the quality of the microbial community against the degradation of the organic pollutant load; discussing these characteristics before starting the discussion of the nitrogen removal process is suggested.
10. In Figure 4B, why is the nitrate concentration lower than that of nitrite, mainly on day 74, where the nitrate concentration would be expected to be higher as a result of the nitrification process? It is suggested to include in this graph the NH4 concentration to observe the nitrification process, and it is also important to associate the biomass concentration, as was the behavior during the whole measurement process. Although the initial concentration is indicated, given that the degradation process is not only associated with bacteria but also with cyanobacteria, microalgae, fungi, etc., it is suggested to include values of Volatile Suspended Solids.
11. Lines 334 to 342 have a different style format than the one indicated in the journal.
12. The results of Figure 6, about the predominance of microbial communities, it is suggested to include a discussion of the population variations with the variation of the physicochemical characteristics of the wastewater.
13. In Figure 7, it is striking that the concentration of nitrate is very low; it is suggested to add a discussion on this aspect contrasting the microbial communities present; it would be expected that the concentration of nitrate is higher if the given process is nitrification and it is an aerobic process. An ANAMMOX? The process is presented, and it is suggested to review the nitrogen cycle and biological nitrogen fixation to discuss the interesting results of the work.
Author Response
No reply.
Below you can find the letter to Assistant Editor.
Dear Ms. Lin,
We are writing you to express our concern regarding the management of the editorial process for our manuscript microorganisms-2144116.
We have received the first two reviewer reports the 2nd of January.
Then the third report, the 4th of January.
Finally, we received the fourth report the 6th of January, but it was communicated the 8th of January, while you are providing as re-submission deadline the 12th of January.
While we were able to address all the concerns expressed by the first three reviewers, the requested revisions from the fourth reviewer cannot be addressed in such short notice. Actually, her/his comments would require an extensive revision of the manuscript, since wet-lab experiments have been requested (2-3 months of additional work).
I would kindly ask you to consider only the first three reviewers; otherwise we are forced to withdraw the paper.
Please let us know as soon as possible.
Best regards,
Maurizio Falconi
Reviewer 3 Report
The present study may add to existing knowledge and be a reference for further studies. Comments are addressed below for manuscript improvement.
Major comments
- In the abstract, the discussed ideas in the sentence “The analysis of data...” is unclear. In this sense, authors are advised to move and replace the last paragraph of the introduction section with the abstract (lines 25-31). Besides, it needs to highlight the implications of this research for the state-of-science and/or practical application in the abstract.
- Lines 48-51: Please, highlight that this classification is mostly valid for landfills located in temperate climatic conditions, different from leachates generated in tropical areas. You may ckeck: 10.1016/j.jenvman.2021.112475; 10.1177/0734242X221116212. Besides, authors are advised to expand the work's background and include the main treatments used to remove ammonia nitrogen (e.g., air stripping, struvite precipitation, biological systems, and ion exchange). Recommended refs not limited to those: 10.31025/2611-4135/2020.13897; 10.1007/s11356-020-10397-9; 10.1016/j.jwpe.2020.101572; 10.1080/10934529.2022.2101842.
- Introduction: Please, highlight the novelty (compared with previous works) and significance of this study.
- Item 2.1: Please, expand this section identifying the type(s) of physicochemical treatment used before leachate is sent to the WWTP. Is desinfection requerired by Italian legislation? Authors should further clarify why low-biodegradable leachate (BOD/COD = 0.3) is treated by biological systems.
- It is suggested to include Italian or European discharge standards besides values identified in table 1.
- Please, include references in items 2.2 and 2.3.
- Section 3.1: lines 196-217 should be removed. Repetitive pieces of information that could be added in section 2.1 succinctly.
- Lines 229-231: references?
- Section 3.4 should be spritted in a research prospects section. A conclusion section must be added to the manuscript.
Minor comments
- Please, clarify the meaning of FM, Italy in the abstract.
- Please, use the international system of units (e.g., meters (m), tonnes (t), etc.)
- Since it was not reported quantitative values (e.g., colour unit), authors are advised to remove colour and smell parameters from table 1.
- Line 108: Would it be anoxic conditions?
- What LMM means?
- Fig. 3B: correct the axis (i.e., NH4+-N)
- The quality of fig. 5 should be improved.
Author Response
We thank the Reviewers and Editors for the positive evaluation of our research and for their valuable suggestions to improve the manuscript. We hope that it is now suitable for publication in Microorganisms.
All changes in the manuscript are indicated as follows:
Deleted text = struck through words
New text, corrections = red words
Reviewer 3
The present study may add to existing knowledge and be a reference for further studies. Comments are addressed below for manuscript improvement.
Major comments
- In the abstract, the discussed ideas in the sentence “The analysis of data...” is unclear. In this sense, authors are advised to move and replace the last paragraph of the introduction section with the abstract (lines 25-31). Besides, it needs to highlight the implications of this research for the state-of-science and/or practical application in the abstract.
Answer. In our opinion, results from the NGS identifying bacteria responsible for ammonia removal cannot be removed from the Abstract. We now report in the Conclusions the implications of our work.
- Lines 48-51: Please, highlight that this classification is mostly valid for landfills located in temperate climatic conditions, different from leachates generated in tropical areas. You may ckeck: 10.1016/j.jenvman.2021.112475; 10.1177/0734242X221116212.
Answer. This relevant point has been added at pag. 2.
- Besides, authors are advised to expand the work's background and include the main treatments used to remove ammonia nitrogen (e.g., air stripping, struvite precipitation, biological systems, and ion exchange). Recommended refs not limited to those: 10.31025/2611-4135/2020.13897; 10.1007/s11356-020-10397-9; 10.1016/j.jwpe.2020.101572; 10.1080/10934529.2022.2101842.
Answer. This point and relative citations have been added at pag. 2.
- Introduction: Please, highlight the novelty (compared with previous works) and significance of this study.
We now report in the Conclusions the novelty and significance of this study.
- Item 2.1: Please, expand this section identifying the type(s) of physicochemical treatment used before leachate is sent to the WWTP. Is desinfection requerired by Italian legislation? Authors should further clarify why low-biodegradable leachate (BOD/COD = 0.3) is treated by biological systems.
Answer. Physicochemical treatments used have been added in section 2.2 and an explanation provided.
- It is suggested to include Italian or European discharge standards besides values identified in table 1.
Answer. Italian limits for ammonia, nitrite and nitrate have been reported in the Introduction (pag. 2).
- Please, include references in items 2.2 and 2.3.
Answer. Procedures reported in sections 2.2 and 2.3 are carefully detailed. In addition, official methods are used at Eco Control Laboratory (Certificated Laboratory, UNI CEI EN ISO/IEC 17025:2018).
- Section 3.1: lines 196-217 should be removed. Repetitive pieces of information that could be added in section 2.1 succinctly.
Answer. Section 3.1 has been shortened removing repetitive sentences.
- Lines 229-231: references?
Answer. Cl- concentration and other parameters are routinely determined at the Eco Control Lab.
- Section 3.4 should be spritted in a research prospects section. A conclusion section must be added to the manuscript.
Answer. We added a conclusion section accordingly.
Minor comments
- Please, clarify the meaning of FM, Italy in the abstract.
Answer. FM indicates the province of Fermo
- Please, use the international system of units (e.g., meters (m), tonnes (t), etc.)
Answer. Corrections have been done.
- Since it was not reported quantitative values (e.g., colour unit), authors are advised to remove colour and smell parameters from table 1.
Table 1 has been changed according to reviewer advise.
- Line 108: Would it be anoxic conditions?
Answer. Anoxia has been changed with anoxic.
- What LMM means?
Answer. LMM indicates the Leachate Minimal Medium and its composition is reported in section 2.2.
- Fig. 3B: correct the axis (i.e., NH4+-N)
Answer. Figures 2B and 3B have been corrected.
- The quality of fig. 5 should be improved.
Answer. Figure 5 has been replaced with a better image.
Reviewer 4 Report
1. In introduction, some relevant studies done by other scholars can be added
10.1016/j.jclepro.2022.130479
10.1016/j.scitotenv.2022.156124
10.1016/j.scitotenv.2022.158254
2. The text formatting in line 172 affects the reading and is suggested to be adjusted.
3. The note content of the chart in line 234 is too long, and the later charts have similar problems.
4. Figure 5 is not clear.
5. In chapter 3, the authors only give a detailed description of the experimental results and lack more discussion and analysis.
6. A chapter summarizing the conclusions of the experiments is missing.
Author Response
We thank the Reviewers and Editors for the positive evaluation of our research and for their valuable suggestions to improve the manuscript. We hope that it is now suitable for publication in Microorganisms.
All changes in the manuscript are indicated as follows:
Deleted text = struck through words
New text, corrections = red words
Reviewer 4
- In introduction, some relevant studies done by other scholars can be added
Answer. According to Reviewer request, the review entitled “Resource utilization of municipal solid waste incineration fly ash………” (doi: 10.1016/j.scitotenv.2022.158254) has been added. In the Introduction has been explained the situation of waste incineration in Italy. Notably, our work is focused on leachate treatment.
- The text formatting in line 172 affects the reading and is suggested to be adjusted.
Answer. Resolved, accordingly.
- The note content of the chart in line 234 is too long, and the later charts have similar problems.
Answer. We disagree with this comment. We feel that the legends are adequately reported.
- Figure 5 is not clear.
Answer. Figure 5 has been replaced with a better image.
- In chapter 3, the authors only give a detailed description of the experimental results and lack more discussion and analysis.
Answer. We now provide a new section, as Conclusions, providing more discussion about the results.
- A chapter summarizing the conclusions of the experiments is missing.
Answer. We defined a new chapter summarizing the conclusions of the experiments
Round 2
Reviewer 1 Report
I accept the text in its present form and recommend its publication.
Author Response
We thank the referee for His/Her constructive criticisms and suggestions
Reviewer 2 Report
Most of the comments were considered, although others were not incorporated. In Table 1, where the results of the physicochemical characterization are presented, it is suggested that the methods used for the measurement of each parameter be included; likewise, since they are averages, it is recommended that the standard deviation of the data be included.
Figures 6 and 7 suggest that the results be discussed by contrasting the physicochemical characterization at the time of sampling for each of the isolates.
Author Response
We apologize to the Reviewer 2 for the unpleasant inconvenience occurred in the first round of revision but we saw his/her comments only January the 9th. We thank the Reviewer for his/her patience and for giving us the chance to address some important points in the second round of revision.
Most of the comments were considered, although others were not incorporated. In Table 1, where the results of the physicochemical characterization are presented, it is suggested that the methods used for the measurement of each parameter be included; likewise, since they are averages, it is recommended that the standard deviation of the data be included.
Answer. Methods used have been added to Materials and Methods section and standard deviation is now included in Table 1.
Figures 6 and 7 suggest that the results be discussed by contrasting the physicochemical characterization at the time of sampling for each of the isolates.
Answer. The isolates (Fig. 7) have been taken from RRIA experiment between the 53th and 75th day (now reported in section 3.4) and tested for their activity in NH4 degradation. The RRIA experiment (Fig. 4) is based on repetitive re-inocula/dilutions, of the same culture, with fresh leachate medium. At each re-inoculum, 30 mL of LMM were added to 10 mL of culture thus restoring the starting condition (ammonia conc. peaks, panel A). Notably, NO2 and NO3 conc. was measured at ammonia falls (panel B). Thus, we can consider the physiochemical situation as stable across the different time points of RRIA except for the number of ammonia stresses occurred. Since bacteria of the sludge were growing slowly in LMM broth, biomass obviously progressively decreased because of numerous dilutions in RRIA.
The starting growth conditions of isolates (Fig. 7) are the same of RRIA (LMM broth). Cells were slowly growing and the optical density increased from 0.1 to 0.3-0.4 after 12 days experiment. Now this information has been reported in the legend of Fig. 7. Unfortunately, we did not measure the progressive change of the physiochemical parameters as BOD and COD. We will take into consideration this suggestion for the progress of this study. Now we acknowledge this limitation in the discussion (please see pag. 12). In addition, a correlation at family level between isolates and NGS data (Fig. 6A) has been added (please see pag. 12).
Reviewer 3 Report
Please, in the abstract, elaborate on the practical significance of results.
Please, include Cl- concentration in the table S1.
Author Response
Please, in the abstract, elaborate on the practical significance of results.
Answer. A sentence has been added at the end of the Abstract.
Please, include Cl- concentration in the table S1.
Answer. Cl- values have been added in table S1